# Autism Spectrum, Hikikomori Syndrome and Internet Gaming Disorder: Is There a Link?

**DOI:** 10.3390/brainsci13071116

**Published:** 2023-07-23

**Authors:** Liliana Dell’Osso, Giulia Amatori, Dario Muti, Federico Giovannoni, Francesca Parri, Miriam Violi, Ivan Mirko Cremone, Barbara Carpita

**Affiliations:** Department of Clinical and Experimental Medicine, University of Pisa, 56126 Pisa, Italy

**Keywords:** hikikomori, autism spectrum, gaming disorder, video game use, autistic traits, social withdrawal

## Abstract

The aim of this study is to review the available literature investigating the relationship between hikikomori, a pathological condition characterized by severe social withdrawal or isolation, autism spectrum disorder (ASD) and Internet gaming disorder (IGD). Studies on the relationship between ASD and IGD have found significant positive correlations between these two conditions. Individuals with ASD would appear to be at risk of developing a problematic use of the Internet, which, to the right extent, would represent a useful tool for social interaction and cognitive development. Even subjects with hikikomori, in whom rarefied interpersonal relationships and social isolation could be balanced by the use of online connections, appear to be at high risk of developing IGD. On the other hand, the finding of significant autistic traits in populations with hikikomori could lead to considering this psychopathological condition as a particular presentation of autism spectrum, a hypothesis that requires further investigation.

## 1. Introduction

Autism spectrum disorder (ASD) is a neurodevelopmental disorder characterized by persistent deficits in social communication and a restricted, repetitive pattern of behaviors, interests, or activities, caused by both genetic and environmental factors affecting the developing brain [1,2]. ASD is now recognized to occur in 1 in 36 children [3]. While comorbidity with other mental disorders is not an uncommon occurrence [1], the scientific literature stresses how milder forms of ASD, without intellectual or language impairment, may remain under-recognized and come to clinical attention after the development of other comorbid psychiatric symptoms, often leading to a misdiagnosis in this kind of patients [4,5,6,7]. From social anxiety disorder to obsessive compulsive and mood disorders, several conditions may mask an ASD among high-functioning adults [8,9,10,11]. As a consequence, the need to carefully evaluate symptoms of the autism spectrum and their possible overlaps with other kinds of mental disorders has been increasingly reported [12]. On a side note, several authors have also highlighted that not only full-fledged ASD conditions but also subthreshold autistic traits, such as those reported among first-degree relatives of ASD subjects can exert a significant impact on the quality of life, being also a significant risk factor for the development of other psychiatric conditions, and thus they should be carefully evaluated and addressed in several kinds of patients [5,6,7,8,9,10,11,12].

In this framework, the recent literature has reported a possible association between autism spectrum features and a specific psychopathological condition called “hikikomori” [13].

Hikikomori is a form of pathological social withdrawal or social isolation lasting longer than 6 months, associated with a significant functional impairment or distress [14,15]. This condition, firstly identified and addressed in Japan, has received increasing attention since the late 1990s [16,17]. Subsequently, several cases have been described in other countries such as South Korea, Spain, and the United States, making hikikomori a well-known pathological condition in the scientific community, worthy of global attention [18,19,20,21,22,23,24,25,26]. Despite that, to date, since hikikomori has not yet been included in the categorial system of the DSM-5-TR [1], where it is described only as a culture-bound syndrome, there is still a lack of consensus about diagnostic criteria. The currently most common definition contains the following three key elements: (1) marked social isolation in one’s home, which includes the physical withdrawal in one’s own place of residence, the absence of active participation in academic and labor settings, as well as the limited involvement in social relationships; (2) the duration of the social isolation is at least six months; (3) the social isolation is associated with significant functional impairment or distress [27]. In previous years, one of the main controversies in this field was the possible link of hikikomori syndrome to a specific cultural environment, with authors questioning whether we could speak of hikikomori outside Japan: in particular, hikikomori syndrome was believed to be associated to specific sociocultural stressors, such as the need to respond to high social requests that some subjects of the new generation feel unable to fulfill [28]. However, the condition to date has been widely observed in several different countries, while similar social stressors have also been reported among young people in most of the industrialized world, partially linked to the crisis of the economic environment of the last decades and the increasing prevalence of the predisposing condition of “not in Employment, Education or Training” (NEET) [29]. In order to better stress how this condition can occur in different cultures and environments, some authors proposed to rename it as “Extreme Severe Withdrawal” (ESW) rather than hikikomori. This point of view was supported by the increasing reports of hikikomori condition from countries outside Japan, such as South Korea, India, Australia, Bangladesh, Iran, Taiwan, Thailand, USA, Oman, France, Italy, Spain, Brazil, Hong Kong, China, and Canada [19,20,30,31,32,33,34,35].

Hikikomori is known to be comorbid with various mental disorders including anxiety disorder, depression, personality disorders, and specifically, Internet gaming disorder (IGD) [23,36,37,38]. IGD is a condition that was recently included in “Section III” of the DSM-5-TR within the chapter “conditions for further studies” and defined as “the persistent and recurrent use of the Internet for gaming, leading to a clinically significant impairment” [1]. Diagnostic criteria for IGD feature an excessive preoccupation with gaming, an increased time spent playing games before feeling satisfied, withdrawal symptoms, compromised relationships, job, or education due to the gaming activity, and others. From a psychopathological point of view, this phenomenon has been associated by numerous studies to the category of substance use or behavioral addictions, with which IGD has different common features, such as mood alterations, social withdrawal, intra- and interpersonal conflicts, relapses, and tolerance [39]. Furthermore, IGD has been associated with a reduction in self-esteem, social skills, social acceptance, promoting a poorer quality of life and dysfunctional coping reactions [40]. Adolescents and young adults seem to be at a higher risk of developing IGD, and some game dynamics such as massively multiplayer online games (MMOs) appear to promote the development of addiction to online gaming [41]. The relationship between hikikomori and IGD have already been highlighted in the literature, with authors noting how one condition may facilitate the development of the other in a vicious circle, to the point that the association between IGD (and more generally an Internet addition) and hikikomori was compared to a “chicken and egg dilemma” [15]. In particular, hikikomori subjects would typically isolate themselves from the outside world, preferring to spend all their time in their own room. This condition seems to facilitate the use of the Internet as the only tool for maintaining some relationships and, on the other hand, an approach to the virtual world, and in particular videogames, oriented to escaping and, in some sense, compensating for the lack of real-life social interactions. On the other hand, the development of an addiction to Internet gaming may promote and maintain the tendency to social withdrawal [42].

This association was also confirmed by experimental studies. In two large samples of young adults evaluated for the presence of IGD and Hikikomori, an association was reported between symptoms of the two conditions, especially when longer game-playing times were implied [43,44]. Moreover, about half of hikikomori patients seem to be at risk of developing an Internet addiction, and about one-tenth meet the diagnostic criteria for this condition [45].

In this regard, it should be noted that one of the populations at greatest risk of isolation due to difficulties in communication and social interaction is individuals with ASD. However, studies about hikikomori and autism are still very scant. Hikikomori has been found to co-occur in around a third of people diagnosed with ASD [46], who are themselves at higher risk of developing an Internet addiction. Young individuals with ASD are indeed reported to frequently overuse video games [47,48,49], spending approximately 62% more time watching television and playing video games than all nonscreen activities combined. In addition, compared with typically developing peers, ASD people tend to spend more hours a day playing video games [50]. It was hypothesized that the greater attention to detail, weak central coherence, and the visual skills sometime showed by people with ASD may make them particularly adjusted to video game tasks requiring rapid visual scanning for stimuli in a complex environment [51]. In addition, children with ASD face many challenges in their daily life and school settings, possibly living the challenges presented in video games as more manageable, enjoyable, or rewarding than other tasks that are presented during the day [52,53]. Video games could even enable these subjects to circumvent their difficulties through imaginative play and improve executive functions [54]. While computer game habits in individuals with ASD might be difficult to distinguish from restricted interests and repetitive behaviors [50], playing video games may also be progressively felt as indispensable, becoming associated with a clinically significant distress, and eventually addiction-related features, in the long term.

Despite the above-reported evidence and the theoretical hypothesis about how autism spectrum, IGD, and hikikomori seem to be linked, studies about the relative relationships between these conditions are still limited in number, while no study have yet investigated the association between autism spectrum, IGD, and hikikomori in the same sample. In order to pave the way for further research in the field, the aim of the present work was to review all available studies about the relationship between autism spectrum and hikikomori, hikikomori and IGD, IGD and autism spectrum, highlighting possible overlapping features and traits.

Furthermore, considering that within the general population, the characteristics of autism are variably distributed along a continuum of severity, from subthreshold manifestations or autistic traits (ATs) to full-blown disorder (ASD), it would be interesting to explore the presence of ATs in individuals with IGD and hikikomori, through the use of specific spectrum questionnaires, such as the Adult Autism Subthreshold Spectrum (AdAS Spectrum).

### 1.1. Hikikomori and ASD

The interest in the possible relationship between autism spectrum conditions and hikikomori has increased only recently.

In the late 2000s, Kondo and colleagues highlighted the presence of developmental disabilities in at least about a third of hikikomori subjects. The authors also stressed that the association with neurodevelopmental disorders should not surprise, considering that subjects with aloof personality, social difficulties, and withdrawal could have likely be diagnosed with autism spectrum conditions before the development of the hikikomori construct [55]. Tateno et al. [38] evaluated a total of 1038 patients diagnosed with hikikomori, showing that up to one-fifth could be diagnosed with a pervasive developmental disorder (PDD) according to the criteria of the ICD-10 [56]. In support of this finding, another study showed that 31.8% of the 183 hikikomori patients followed by mental health centers had developmental disorders such as mental retardation and PDD [55], according to DSM-IV-TR criteria [57]. A more recent study evaluated instead the correlation between autism spectrum and hikikomori in a sample of 416 clinical patients recruited through the Mood Disorder/Hikikomori Clinic. Among all the cases, 103 hikikomori and 221 clinical controls without the hikikomori condition were extracted using a semistructured interview and were requested to fill out a set of self-rated scales, including the Japanese version of the Autism-Spectrum Quotient (AQ-J) [13]. Compared to healthy controls, subjects diagnosed with hikikomori reported a higher score on the Autism-Spectrum Quotient—Japanese version (AQ-J), showing lower social skills, lower communication and imagination abilities, a lower attention span, a lower multitask ability, and a lower adaptation to change: these findings suggest a higher prevalence of autistic traits in hikikomori than in the general population [13]. Shimono et al. [58] used the AQ on a sample of 272 university and graduate students with the aim to evaluate whether the presence of autistic traits could be predictive of the “affinity for hikikomori”, a precursor condition of the hikikomori syndrome [58,59,60,61]. Results showed that autistic traits, measured with the Autism-Spectrum Quotient (AQ), and especially difficulties in social interaction, were predictors of the maladaptive aspect of the hikikomori affinity. The affinity for hikikomori scale in university students was used for assessing the two dimensions of hikikomori tendencies: the desire for hikikomori, and an empathy for others with hikikomori. The desire for hikikomori subscale and empathy for others with hikikomori subscale feature 10 and 6 items, respectively. In that study, the instructions contained in the original Affinity of Hikikomori Scale [62] were also followed. Finally, Brosan et al. [46] showed that, in a sample of 646 people aged 16–24, the relationship between psychological wellbeing and hikikomori risk was mediated by autistic traits, and that individuals with higher levels of autistic traits who did not leave the house were at higher risk of hikikomori. Another 2023 study led by Yamada et al. evaluated 39 adult male patients diagnosed with ASD. Subjects were assessed through structured interviews such as the AQ, the Autism Diagnostic Observation Schedule (ADOS), and the Fibromyalgia Impact Questionnaire (FIQ) and were subsequently divided into two groups: ASD with hikikomori condition (N = 16) and ASD without hikikomori condition (N 23). That study found that compared to non-hikikomori controls, hikikomori cases had stronger sensory symptoms, lower uric acid, higher rates of atopic dermatitis, more severe depressive and social anxiety symptoms based on self-reported scales such as the Patient Heath Questionnaire 9 (PHQ-9), the Liebowitz Social Anxiety Scale Japanese Version (LSAS-J), and the Modern-Type Depression Trait Scale (TACS-22) [14,63]. The correlation between hikikomori and modern-type depression had already been highlighted in the literature [13,64]. These studies are summarized in Table 1.

### 1.2. Hikikomori and IGD

Although the close relationship between these conditions have been variously described in theoretical studies [15], data from the literature are still limited. The first relevant study in the field was conducted in Japan [65] on a sample of 478 university students from private universities in the Sapporo region, aged between 19 and 20 years (132 males, 355 females). The study aimed to investigate the relationship between Internet use and hikikomori traits in the young adult population of Japan, through the use of a set of questionnaires for analyzing the subjects’ type of Internet use such as the Young’s Internet Addiction Test (IAT), the Smartphone Addiction Scale (SAS)—Short Version, the 25-item Hikikomori Questionnaire (HQ-25), the Tarumi scale on Modern Depression called Avoidance of social roles, Complaint and low Self-esteem (TACS) [14]. Time spent on the Internet was found to be two hours longer on weekends than on other days of the week, without significant differences between genders; the main online activities were social networking, video sharing, and video gaming. Furthermore, the use of the Internet for gaming purposes was extremely higher in men (18.9%) than in women (4.8%), while smartphone use was drastically higher in females than in males. Regarding hikikomori traits, 22% of male subjects fell into the category at risk for hikikomori, according to the HQ-25 score.

Individuals at risk for hikikomori were also found to have longer internet usage time and higher scores on both IAT and SAS. Moreover, correlation analyses revealed that HQ-25 and IAT scores had a relatively strong relationship, although HQ-25 and SAS-SV had a moderately weak one. Lastly, those who used the Internet for gaming were at higher risk for the development of hikikomori than those who used the Internet for social networking. According to the authors, these data could be possibly explained by the fact that gaming, especially when competitive, is an activity better performed on the computer than on the smartphone, inducing the subject to spend more time in their room to have a better gaming experience. However, this may enhance a reduction of outside social activities. Another study of interest was carried out in the USA and Australia [44] on two samples of young habitual online gamers (18–29 years old, 153 from Australia and 457 from the USA). These subjects were specifically selected as MMO players, a particular type of online game characterized by the possibility to be connected to hundreds or thousands of players simultaneously, typically set in a gigantic persistent virtual world. The aim of the study was to investigate the correlation between hikikomori and IGD and to analyze symptoms suggestive of hikikomori as potential risk factors for IGD. The information collected on the subjects included the number of hours they spent playing games each day and whether the subject lived with their parents. The tools used were the Internet Gaming Disorder Scale—short form and the Hikikomori Social Withdrawal Scale (HSWS) [23]. Results showed that subjects with more pronounced hikikomori symptoms also reported significantly higher scores on the IGD scale, both in American and Australian subjects, an indication that the different nationality did not affect the survey. It was also observed that each additional hour of videogame use increased both the risk of developing IGD and hikikomori symptoms. These studies are summarized in Table 2.

### 1.3. ASD and IGD

ASD people have often been reported to show a problematic and excessive video game use, spending most of their free time on screens, social media, television, or computers [51]. Orsmond et al. [51] led an investigation on 406 families of adolescents and adults with ASD aged between 12 and 21 years. All patients met the criteria for ASD according to the Autism Diagnostic Interview-Revised (ADI-R). Of the 393 families who continued to participate in the study, 145 included an adolescent (ages 12–21) with an ASD subject who lived in the house. The mothers of 103 teenagers with ASD completed two 24-h diaries to describe participation in the activity of their sons, for a response rate of 71%. It turned out that the evaluated teenagers spent a lot of time on solitary activities, especially on screens and using computers and television, much less on activities with peers. Another study from Mazurek and Wenstrup [48] analyzed a sample of 202 children and adolescents with ASD (age 8–18) compared to typically developing siblings (TD) (*n* = 179). Parents completed the measures of evaluation of extracurricular activities and on the screen of children. Children with ASD spent about 62% more time watching TV and playing video games than all activities not on the screen. Compared to children with typical development, it was shown that children with ASD played video games more frequently and were therefore more likely to demonstrate a problematic use of video games, including difficulties in stopping the gaming activity when necessary, anger when interrupted, and excessive amounts of play, with no gender differences [48]. The same study showed that most children with ASD play video games in a solitary way, thus not resulting in an improvement of their social skills. Compared to the typical development sample, children with ASD spent more hours a day playing video games (2.4 versus 1.6 for boys and 1.8 versus 0.8 for girls) and had higher levels of problematic video game use. In contrast, children with ASD spent little time using social media or socially interactive video games. Parents filled out a demographic and history form to provide information on child and family variables. Information included age, race, parental marital status, family income, number of siblings, child with ASD diagnostic information, child with ASD diagnostic information, parent-reported IQ information (if known), and specific information about activity and on media usage based on the screen. A modified version of the Problem Video Game Playing Test (PVGT) was used to examine problematic aspects of the game.

Mazurek and Engelhardt [47] conducted a study on parents of children/adolescents with ASD (*n* = 56), attention deficit hyperactivity disorder (ADHD) (*n* = 44) and controls (*n* = 41) aged between 8 and 18 years (mean 11.7 years, SD 2.5 years). A significant difference between the ASD group and controls was found on the daily hours of video games played, the family income, and the marital status of parents. Results suggested that the ASD group had higher PVGT scores than controls. However, symptoms of GD were not related to autistic traits in the ASD group. The ASD diagnosis was assessed clinically by a physician and/or psychologist and by the use of standardized tools, including the ADOS or ADI-R. The questionnaires assessed hours of video game use per day, access to in-room video games, video game genres, problematic video game use, ASD symptoms, and ADHD symptoms. Boys with ASD spent more time than controls playing video games (2.1 vs. 1.2 h/d). Both the ASD and ADHD groups had greater access to in-room video games and more problematic use of video games than controls. Symptoms of inattention predicted problem gambling for both the ASD and ADHD groups, and role-playing preferences predicted problematic game use only in the ASD group. Video game use was assessed using a questionnaire specifically designed for the study. Parents reported the number of hours per day their children spent “playing videos or video games” during extracurricular hours. Parents also listed the top three most played video games by their kids in the past month. Problematic video game use was assessed using a modified version of the PVGT. Current ASD symptoms were assessed using the Social Communication Questionnaire-Current (SCQ). Symptoms of inattention and hyperactivity/impulsivity were assessed using the Vanderbilt Attention Deficit/Hyperactivity Disorder Parent Rating Scale (VADPRS).

MacMullin et al. [66] evaluated a sample of 172 parents of typically developing youth and 139 parents of youth diagnosed with ASD between 6 and 21 years of age. All parents completed the Social Communication Questionnaire (SCQ) about their offspring. Parents were asked if their child knew how to use electronic devices and the electronic activities performed, frequency of use, age of onset of use, amount of time spent. Parents were asked if their child’s use of electronics was currently causing problems for them or their child in any way. Problematic Internet use was assessed using the Compulsive Internet Use Scale (CIUS). Problematic video game use was measured by replacing “Internet” with “video game” in the CIUS. Individuals with ASD were reported to have used certain electronic devices more often in the past month and on an average day, and to show a greater compulsive use of the Internet and video games than individuals without ASD. In both samples, boys played video games more often than girls. Compared with parents of individuals without ASD, parents of individuals with ASD were significantly more likely to report that electronics use was currently having a negative impact.

In a further study, Arcelus et al. [67] investigated the relationship between GD and autistic traits in a sample of 245 subjects from a national transgender service in the United Kingdom (UK). Among them, 154 (62.9%) individuals were self-reported current players. Participants were asked to self-report information about whether or not they were engaging in gambling behavior (yes or no) and whether they had engaged in gaming behavior in the past (yes or no). This information was used to classify participants as “current players” or as “non-players”. Participants were also assessed with The Internet Gaming Disorder Scale (IGDS9-SF), Interpersonal Problem Inventory (IIP-32), Hospital Anxiety and Depression Scale (HADS), and the Autism Spectrum Quotient 28 (AQ-28). Results highlighted that gamers more frequently showed a younger age, interpersonal problems, depression, and higher scores on the AQ-28.

Liu et al. [68] instead investigated the relationship between autistic traits and Internet gaming addiction in a sample of 420 Chinese children (mean age 9.74 years). The study featured a longitudinal design, following subjects from the fall of the fourth grade (T1) until the spring of the fifth grade (T4). Participants were assessed with the Social and Communication Disorders Checklist (SCDC) as a measure of autistic traits. Internet gaming addiction was measured by adapting the Pathological Video Game Use Questionnaire. Subjects also fulfilled the Emotion Regulation Questionnaire (ERQ) and the Emotional Engagement subscale of the School Engagement Scale. Results showed significant positive correlations between autistic traits and gaming addiction, while emotional regulation and school connectedness reported significant negative correlations with both autistic traits and gaming addiction. Moreover, the authors reported a significant longitudinal effect of autistic traits in predicting Internet gaming addiction at T4. The effect was both direct and indirect through the prediction of a reduced emotional regulation and school connectedness at T2 and T3, respectively. Globally, the model supported the potential effect of autistic traits in promoting the development of gaming addiction.

Similarly, Engelhardt [69] and Paulus [49] reported that subjects with ASD were more likely to demonstrate increased symptoms of GD in comparison to controls. The study by Engelhardt et al. [69] evaluated whether adults with ASD were more at risk of GD than healthy adults. Participants included 119 adults (16 women) with and without ASD. The ASD group were recruited through a diagnostic process that included structured interviews, behavioral observations, professionally conducted assessments using assessment tools such as the ADOS and the ADI-R. Adults with ASD were more prone to pathological video game use than controls, with more hours per day engaged in video game activities and a higher percentage of leisure time used. Participants completed the measurements by assessing hours of video game use per day, percentage of free time spent playing video games, and pathological game use symptoms. The study of Paulus et al. [49] investigated children and adolescents with ASD who played computer games and computer-mediated communication (CMC) compared to their control peers. Parents completed a standardized questionnaire on media use, gaming disorder (GD), and CMC. Sixty-two boys diagnosed with ASD between the ages of 4 and 17 years (mean = 11.5; SD = 3.2) were compared with 31 healthy boys (mean = 11.5; SD = 3.7).

In a study conducted by Finke et al. [53] on a sample of nine males and one female with autism spectrum conditions, aged from 18 to 24 years, the amount of time spent playing video games per week ranged from 7 to 53 h, with an average of about 26.8 h per week and 3.8 h per day. The purpose of this survey was to better understand the experiences of people with ASD playing video games as the main form of leisure through a quantitative research methodology and, in particular, through a semistructured interview given to patients who had the opportunity to describe their experiences and also the benefits of playing video games. The results indicated that participants felt that playing video games had a positive impact on their lives and development. The motivations for playing video games were similar to those reported by typically developing populations. More recently, a systematic review by Murray et al. [70] explored the association between autism and a broad problematic use of the Internet (PIU) as well as GD. A total of 2286 publications were screened, and 21 were considered eligible for inclusion in the review. Most studies found positive associations between PIU and subclinical autistic-like traits with weak and moderate effect sizes and between ASD and PIU with varying effect sizes. Additionally, people with ASD were more likely to exhibit GD symptoms. The authors reported a total of five studies on the relationship between ASD and GD, which all reported significant positive associations [7,47,48,49,66]. The two identified studies about subthreshold autistic traits and GD also reported significant associations [67,68].

Finally, Murray et al. [71] mined whether IGD symptoms were higher among ASD adults (n = 230) compared to a control group (n = 272), also exploring gaming disorder (GD) predictors. The average age of the ASD group was 31.32 years and the average age of the control group was 29.51 years. Measures included the AQ-10, the Ten-Item Internet Gaming Disorder test, the parents and peer attachment inventory, the emotional regulation questionnaire, the Questionnaire on Social Functioning (SFQ) and the NEO Five-Factor Inventory-3 (extroversion facet). As hypothesized, symptoms of GD were significantly higher in participants with ASD than in the control group, with 9.1% of the ASD group and 2.9% of the control group classified as suffering from GD. Peer attachment, emotional regulation, and extroversion significantly predicted GD scores. These studies are summarized in Table 3.

## 2. Discussion

### 2.1. ASD and IGD

The aim of this work was to review the available literature about comorbidity and overlaps among autism spectrum, hikikomori, and IGD. First of all, as anticipated, the literature in the field, although promising, is still scant. To date, the most investigated relationship remains that between ASD and IGD. Studies on the relationship between ASD and IGD included in this review, conducted on samples of children, adolescents, and adults with ASD [47,51,70,71], have found significant positive correlations between these two conditions, stressing the need to deepening the investigation on this matter. Starting from the consideration that technological activities, including video games, represent a common interest among people with ASD [72], possibly supported by a cognitive style characterized by a greater attention to details [51], one of the main issues that remain to be clarified is the distinction between increased interest and the use of (internet) video games among subjects with ASD as part of their pattern of special interests and the risk of IGD. It should be noted that the difference between normal and problematic use of the Internet or games has been questioned by some authors [73,74], both conceptually and methodologically. With challenges in defining normal Internet or game use, it could be difficult to classify problematic use [75]. This factor could be exacerbated among autistic populations, in which gaming could represent a special interest and parents may experience anxiety about media use by their children [72]. Some studies have also found that parents show more positive feelings about media use among nonautistic samples [76]. This potential disparity in parental attitudes may play a role in the large differences in GD observed between the ASD and control groups included in this review. However, while the specific risk of IGD in ASD should be carefully assessed, an increased internet and/or video games use among subjects with ASD should not necessarily be considered detrimental. In particular, the well-known difficulties in communication and social interaction that characterize, as nuclear symptoms, patients with ASD could be partially overcome through the establishment of online connections, which are less limited by emotional, social, and time pressures [77]. From this perspective, some authors even suggested that in individuals with ASD, electronic media may be considered a tool for social interaction [77,78,79], with positive effects on well-being and cognitive development [47,48]. However, according to one of the studies revised herein, most children with ASD seem to prefer playing video games solitarily [47,48], while the population of ASD subjects may still feature, overall, a higher risk of developing IGD, especially due to their impairment in the social sphere [80,81]. In this context, studies have highlighted the importance of the number of daily hours spent on video games, which seems to be associated with the degree of social withdrawal and be directly proportional to the increased risk of developing hikikomori [44]. Another possible explanation of the increased interest in video games (and possibly of a greater vulnerability to the development of IGD) among subjects with ASD has been linked to a possible role of video games in circumventing difficulties experienced in a challenging daily life through imaginative play [54]. In this framework, it should also be noted that videogames have been tested as potential therapeutic tools among subjects with ASD, who have also reported a certain response to virtual environment therapies [82,83,84]. In conclusion, the increased use of video games in the autism spectrum condition should be regarded with a specific attention, and further studies should clarify the boundaries between neutral versus detrimental use of video games in this population, trying to identify possible risk factors for the development of IGD.

### 2.2. Hikikomori and IGD

Regarding the association between hikikomori and IGD, the limited available literature suggests the presence of a significant overlap. On one hand, the rarefaction of interpersonal relationships and social withdrawal characteristic of hikikomori may be compensated through the use of the Internet and the online connections established through it [42]. The use of the Internet and online video games as a compensatory technique to interact with others would seem to become pathological in some individuals, leading to the development of a full-blown mental IGD [85]. Among different kinds of Internet use, the MMO game dynamic seems to be the one most associated with the development of hikikomori. These data may be linked to the fact that this game mode is greatly time-consuming and allows the subject to create a more complex net of social relationships in which they could be immerged the entire day and for multiple days on, without the need to meet other people in real life, perfectly pandering to the social needs of hikikomori subjects [86]. Hikikomori subjects also seem to prefer cyber-relationships and, in general, show greater satisfaction from online relationships than from real ones [42]. It has also been observed that in the absence of adequate treatment, hikikomori and IGD would mutually reinforce each other over time, with progressive increases in the severity of both disorders [87]. In particular, one of the main questions related to the link between hikikomori and IGD is which condition would be more likely to start first, then promoting the other, which, in turn, would be reinforced [51]. Indeed, according to some authors [15], both possibilities could be true. While in the DSM-5 TR, where hikikomori is described among the culture-bound syndromes, a possible specific association with IGD is also reported, further studies should clarify the specific boundaries between the two conditions, and specific correlates of hikikomori subjects with higher or lower risks of developing IGD and vice versa.

### 2.3. ASD and Hikikomori

While, as reported above, both ASD and hikikomori subjects seem to be at greater risk of developing IGD, these two conditions also seem to significantly overlap. Despite the limited studies available, since the early investigations on psychiatric conditions among hikikomori subjects, a significant association with developmental disabilities has been reported [15]. Further studies have invariably confirmed the presence of increased autistic traits among the hikikomori population. In the framework of a spectrum approach [88,89], and considering the continuously distributed presence of subthreshold autistic traits from the general to the clinical population, as well as the role of autistic traits in shaping a greater vulnerability towards psychiatric symptoms [12,88], the difficulties in social interaction and communication typical of the autism spectrum could be the basis of the tendency towards a withdrawal among hikikomori subjects [13]. Autistic traits, and especially difficulties in communication and social interaction, seem in fact to be predictors of the maladaptive component of so-called “hikikomori affinity,” defined as the desire to be isolated, a precursor to the full-fledged hikikomori condition [58]. This hypothesis is further supported by one of the last investigations in the field, which highlighted that the presence of autistic traits also seemed to mediate the risk of developing hikikomori [46]. Globally, the investigation on the relationship between hikikomori and autism spectrum should be deepened. Eventually, it should also be clarified whether hikikomori condition could be considered as a peculiar presentation of an autism spectrum, whose development would be linked to specific individual and environmental factors. Investigating in a more comprehensive way the overlapping features between hikikomori, autism spectrum, and IGD may help to find a better conceptualization for the novel constructs of hikikomori and IGD, also allowing researchers and clinicians to shed new light on the autism spectrum psychopathology and its possible emerging presentations in modern times.

## Figures and Tables

**Table 1 brainsci-13-01116-t001:** Main studies on the relationship between hikikomori and ASD.

Kondo [55]	337 individuals with hikikomori (183 subjects of help-seeking group; 154 subjects of non-help-seeking group)	Multiaxial psychiatric diagnosis based on the DSM-IV-TR.	In the help-seeking group 148 subjects had a diagnosis: - 33.3% (49) of the subjects were diagnosed with schizophrenia, mood disorders, or anxiety disorders. - 32.0% (47) of the subjects were diagnosed with developmental disorders. - 34.7% (51) of the subjects were diagnosed with personality disorders, chronic forms of adjustment disorder, or somatoform disorders.
Tateno et al. [38]	1038 patients diagnosed with hikikomori.	Questionnaire regarding hikikomori	- One-fifth could be diagnosed with a pervasive developmental disorder.
Katsuki et al. [13]	416 clinical patients (103 hikikomori, 221 clinical controls without hikikomori)	AQ-J, HAMD-17, BDI-II, TACS-22, Lubben Social Network Scale, Preference for Solitude Scale, Revised UCLA Loneliness Scale, Multidimensional Scale of Perceived Social Support.	- Hikikomori sufferers were more likely to have an autistic tendency. - Hikikomori sufferers with high ASC may have much more difficulty in social communication and social interaction. - Those with high ASC may also have a lower self-esteem and higher complaint tendencies such as aspects of modern-type depression traits, which may relate to the occurrence of hikikomori.
Shimono et al. [58]	272 university andgraduate students	AQ-J, Affinity for Hikikomori Scale (desire for hikikomori subscale; empathy for others subscale), Academic Failure Subscale, Interpersonal Stress Event Scale	- Autistic traits, especially difficulties in social interaction, were predictors of the maladaptive aspect of hikikomori affinity. - Difficulties in social interaction aspects of autistic traits were positively associated with academic failures and a decreased social support from friends. - Social interaction aspects of autistic traits were a factor leading to academic stressors but not interpersonal stressors. - Academic failures were positively associated with the desire for hikikomori
Brosan et al. [46]	646 subjects	NHR, WEMWBS, AQ10, Lockdown Questionnaire.	- The relationship between psychological wellbeing and hikikomori risk was mediated by autistic traits and individuals with higher levels of autistic traits who did not leave the house were at higher risk of hikikomori. - Autistic traits mediated the relationship between both psychological wellbeing and hikikomori risk, as well as between the frequency of leaving the house during lockdown and hikikomori risk. - A greater hikikomori risk was associated with poor psychological wellbeing, higher autistic traits and leaving the house less frequently during the COVID-19 pandemic.
Yamada et al. [63]	39 adult patients diagnosed with ASD (16 with hikikomori condition; 23 without hikikomori condition).	- Self-administered rating scales:AQ, ADOS, FIQ, PHQ-9, LSAS-J, TACS-22. Blood biomarkers: - Serum HDL-cholesterol, LDL-cholesterol, serum total bilirubin, uric acid, high sensitivity C-reactive protein, plasma fibrin degradation products.	- Hikikomori cases had stronger sensory symptoms, lower uric acid, higher rates of atopic dermatitis, more severe depressive and social anxiety symptoms.

ASD = autism spectrum disorder; AQ-J = Autism Spectrum Quotient—Japanese version; HAMD-17 = Hamilton Depression Rating Scale; BDI-II = Beck depression Inventory; NHR = NEET/Hikikomori Risk; WEMWBS Warwick-Edinburgh Mental Wellbeing scale; AQ10 = Autism-Spectrum Quotient—10 items; ADOS = Autism Diagnostic Observation Schedule; FIQ = Fibromyalgia Impact Questionnaire; PHQ-9 = Patient Heath Questionnaire 9; LSAS-J = Liebowitz Social Anxiety Scale Japanese Version; TACS-22 = Modern-Type Depression Trait Scale (TACS-22).

**Table 2 brainsci-13-01116-t002:** Main studies on the relationship between hikikomori and IGD.

Tateno et al. [65]	478 university students	IAT, SAS—Short Version, HQ-25, TACS	- The use of the Internet was about 2 h more on weekends than on weekdays, without significant differences between genders. - The main online activities were social networking, video sharing, and video gaming. - The use of the Internet for gaming purposes was extremely higher in men (18.9%) than in women (4.8%). - Females used smartphones drastically more than males. - 52.4% of the subjects had scored lower than 40 on the HQ-25 (no addiction), 44.4% scored from 40 to 69 (possible addiction), and finally the last 3.3% had higher scores (Internet addiction). There were no significant differences between the sexes. - 22% of male subjects fell into the category at risk for hikikomori. - Gamers showed higher scores on the IAT and HQ-25, as well as a more prolonged use of the Internet itself both in weekdays and on weekends. - Subjects at high risk for hikikomori were those who had significantly higher scores on both the SAS-SV and the IAT and used the Internet longer than low-risk subjects.
Stavropoulos et al. [44]	Two samples of habitual MMO gamers young adults	HSWS; IGD scale—short form	- Subjects with more pronounced hikikomori symptoms were also those who generally had higher scores on the IGD scale; - Each additional hour of videogame use increases both the risk of developing IGD and hikikomori symptoms (the severity of both grows as the hours of play increase).

HQ-25 = 25-item Hikikomori Questionnaire; HSWS = Hikikomori Social Withdrawal Scale; IAT = Young’s Internet Addiction Test; IGD scale—short form = Internet Gaming Disorder Scale—short form; MMO = Massively Multiplayer Online; SAS–Short Version = Smartphone Addiction Scale–Short Version; TACS = the Tarumi scale on Modern Depression called Avoidance of social roles, Complaint and low Self-esteem.

**Table 3 brainsci-13-01116-t003:** Main studies on the relationship between ASD and IGD.

Orsmond et al. [51]	- 103 adolescents with ASD (interviews administered to the mothers)	ADI-R, two 24 h diaries	The evaluated teenagers spent a lot of time in solitary activities, especially on screens and using computers and television, much less in activities with peers.
Mazurek and Wenstrup [48]	ASD: 202 children and adolescents TD: 179	Interview with demographic information PVGT	Children with ASD spent about 62% more time watching TV and playing video games than all activities not on the screen. Compared to children with typical development, children with ASD played video games more frequently and more likely demonstrated a problematic use of video games, including difficulties in stopping the gaming activity when necessary, anger when interrupted, and excessive amounts of play, with no gender differences.
Mazurek and Engelhardt. [47]	ASD: 56 ADHD: 44 TD: 41 (age 8–18)	ADOS ADI-R PVGT SCQ VADPRS	Significant difference between ASD group and controls was found on the daily hours of video games played. ASD group had higher PVGT scores than controls. However, symptoms of GD were not related to autistic traits in the ASD group.
MacMullin et al. [66]	172 parents of TD youth 139 parents of ASD youth	Interview SCQ CIUS	Individuals with ASD were reported to use certain electronic devices more often in the past month and on an average day and to show a greater compulsive use of the Internet and video games than individuals without ASD. In both samples, boys played video games more often than girls. Compared with parents of individuals without ASD, parents of individuals with ASD were significantly more likely to report that electronics use was currently having a negative impact
Arcelus et al. [67]	245 transgender subjects (154 current players 91 nonplayers)	Interview IGDS9-SF IIP-32 HADS AQ-28	Gamers showed more frequently a younger age, interpersonal problems, depression, and higher scores on the AQ-28.
Liu et al. [68]	420 Chinese children	SCDC ERQ PVGT EES	Significant positive correlations between autistic traits and gaming addiction, while emotional regulation and school connectedness reported significant negative correlations with both autistic traits and gaming addiction. A significant longitudinal effect of autistic traits in predicting internet gaming addiction at T4 was found. The effect was both direct and indirect through the prediction of reduced emotional regulation and school connectedness at T2 and T3, respectively. Globally, the model supported the potential effect of autistic traits in promoting the development of gaming addiction.
Engelhardt et al. [69]	119 Adults with and without ASD	Structured interviews, behavioral observations, ADOS ADI-R	Adults with ASD showed more pathological video game use than controls, with more hours per day engaged in video game activities and a higher percentage of leisure time used.
Finke et al. [53]	10 young adults with ASD	Semi-structured interviews of adults who had the opportunity to describe experiences and also the benefits of playing video games.	The amount of time spent playing video games per week ranged from 7 to 53 h, with an average of about 26.8 h per week and 3.8 h per day. The results indicated that participants felt that playing video games had a positive impact on their lives and development. The motivations for playing video games were similar to those reported by TD populations.
Paulus et al. [49]	ASD: 62 boys TD: 31 boys	CMC Interview for parents	The study reported that subjects with ASD were more likely to demonstrate increased symptoms of GD in comparison to controls.
Murray et al. [71]	ASD adults: 230 TD adults: 272	AQ-10 IGDT-10 ERQ IPPA SFQ NEO-FFI-3	Symptoms of GD were significantly higher in participants with ASD than in the control group with 9.1% of the ASD group and 2.9% of the controls classified as suffering from GD. Peer attachment, emotional regulation, and extroversion significantly predicted GD scores.

ADHD: attention deficit hyperactivity disorder; ADI-R: Autism Diagnostic Interview-Revised; ADOS: Autism Diagnostic Observation Schedule; AQ-10: Autism Spectrum Quotient 10; AQ-28: Autism Spectrum Quotient 28; ASD: autism spectrum disorder; CIUS: Compulsive Internet Use Scale; CMC: computer-mediated communication; EES: Emotional Engagement Subscale of the School Engagement Scale; ERQ: Emotional Regulation Questionnaire; GD: gaming disorder; HADS: Hospital Anxiety and Depression Scale; IGDS9-SF: Internet Gaming Disorder Scale; IGDT-10: Ten-Item Internet Gaming Disorder test; IIP-32 Interpersonal Problem Inventory; IPPA: Parents and Peer attachment Inventory; NEO-FFI-3: NEO five-Factor Inventory-3 (extroversion facet); PVGT: Problem Video Game Playing Test; SCDC: Social and Communication Disorders Checklist; SCQ: Social Communication Questionnaire-Current; SFQ: Questionnaire on Social Functioning; TD: typical developing; VADPRS: Vanderbilt Attention Deficit/Hyperactivity Disorder Parent Rating Scale.

## Data Availability

Not applicable.

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
