# Peer review of "Autism Spectrum, Hikikomori Syndrome and Internet Gaming Disorder: Is There a Link?"

_brainsci, 2023, doi:10.3390/brainsci13071116_

Round 1

Reviewer 1 Report

I believe this to be an important paper that integrates 3 supposedly different diagnosis and explores the hypothesis of substantial overlap between them. The introduction is well written and is quite comprehensive. A great review. Table 1 -3 are informative and highly appropriate for the manuscript. This was a large undertaking representing valuable work in this area. There are however a few things that need attention.

On Line 26, the stated prevalence estimate is incorrect. The reference for this (3) is way out of date from 2014. The CDC estimates have recently been published. This needs to be corrected.

The paragraph starting at line 141 is very long. It could be separated into a few paragraphs. The Same with the paragraph at 187 and the discussion.

Several references throughout the paper are inconsistently presented. Some are bracketed and some are smaller font and unbracketed.

none noted

Author Response

Dear reviewer, I write below the response to your correction requests:

1.  Thank you for allowing us to update this data. We have reported the correct percentage as suggested and modified reference number 3. 
2. We have split the longer chapters into paragraphs as suggested. 
3. We have standardized all references in the text, putting those in superscript as a number in square brackets. 

Reviewer 2 Report

The work is current and of particular interest for developmental and clinical psychology. The overall layout is good and the bibliographic sources well identified. 

Author Response

Dear reviewer, thank you very much for your comments.